ථ | Open Peer Review | Microbial Ecology | Observation

# Clearing the plate: a strategic approach to mitigate well-to-well contamination in large-scale microbiome studies

Caitriona Brennan,[1,2] Pedro Belda-Ferre,[1] Simone Zuffa,[3] Vincent Charron-Lamoureux,[3] Ipsita Mohanty,[3] Gail Ackermann,[1] Celeste Allaband,[1] Madison Ambre,[1] Tara Boyer,[1] MacKenzie Bryant,[1] Kalen Cantrell,[4] Antonio Gonzalez,[1] Daniel McDonald,[1] Rodolfo A. Salido,[5] Se Jin Song,[6] Gillian Wright,[1] Pieter C. Dorrestein,[1,3,6] Rob Knight[1,4,5,6]

**ABSTRACT**  Large-scale studies are essential to answer questions about complex microbial communities that can be extremely dynamic across hosts, environments, and time points. However, managing acquisition, processing, and analysis of large numbers of samples poses many challenges, with cross-contamination being the biggest obstacle. Contamination complicates analysis and results in sample loss, leading to higher costs and constraints on mixed sample type study designs. While many researchers opt for 96-well plates for their workflows, these plates present a significant issue: the shared seal and weak separation between wells leads to well-to-well contamination. To address this concern, we propose an innovative high-throughput approach, termed as the Matrix method, which employs barcoded Matrix Tubes for sample acquisition. This method is complemented by a paired nucleic acid and metabolite extraction, utilizing 95% (vol/vol) ethanol to stabilize microbial communities and as a solvent for extracting metabolites. Comparative analysis between conventional 96-well plate extractions and the Matrix method, measuring 16S rRNA gene levels via quantitative polymerase chain reaction, demonstrates a notable decrease in well-to-well contamination with the Matrix method. Metagenomics, 16S rRNA gene amplicon sequencing (16S), and untargeted metabolomics analysis via liquid chromatography-tandem mass spectrometry (LC-MS/MS) confirmed that the Matrix method recovers reproducible microbial and metabolite compositions that can distinguish between subjects. This advancement is critical for large-scale study design as it minimizes well-to-well contamination and technical variation, shortens processing times, and integrates with automated infrastructure for enhancing sample randomization and metadata generation.

**IMPORTANCE**  Understanding dynamic microbial communities typically requires large-scale studies. However, handling large numbers of samples introduces many challenges, with cross-contamination being a major issue. It not only complicates analysis but also leads to sample loss and increased costs and restricts diverse study designs. The prevalent use of 96-well plates for nucleic acid and metabolite extractions exacerbates this problem due to their wells having little separation and being connected by a single plate seal. To address this, we propose a new strategy using barcoded Matrix Tubes, showing a significant reduction in cross-contamination compared to conventional plate-based approaches. Additionally, this method facilitates the extraction of both nucleic acids and metabolites from a single tubed sample, eliminating the need to collect separate aliquots for each extraction. This innovation improves large-scale study design by shortening processing times, simplifying analysis, facilitating metadata curation, and producing more reliable results.

Address correspondence to Rob Knight, rknight@ucsd.edu.

R.K. is a scientific advisory board member and consultant for BiomeSense, Inc., has equity, and receives income. He is a scientific advisory board member and has equity in GenCirq. He is a consultant and scientific advisory board member for DayTwo and receives income. He has equity in and acts as a consultant for Cybele. He is a co-founder of Biota, Inc., and has equity. He is a cofounder of Micronoma, has equity, and is a scientific advisory board member. D.M. is a consultant for, and has stock in, BiomeSense, Inc. P.C.D. is an advisor and holds equity in Cybele, Sirenas, and BileOmix and is a scientific co-founder and advisor for and holds equity in Ometa, Enveda, and Arome with prior approval by University of California San Diego. P.C.D. also consulted for DSM animal health in 2023. The terms of these arrangements have been reviewed and approved by the University of California San Diego, in accordance with its conflict of interest policies.

See the funding table on p. 6.

**KEYWORDS** cross-contamination, well-to-well contamination, large-scale studies, microbiome, metabolomics

Advancements in high-throughput sequencing technologies have enabled researchers to leverage automation and multiplexing, making large-scale studies more cost-effective and efficient. As a result, links between the microbiome and topics ranging from human health to environmental sustainability are revealed nearly every week (1–4). The necessity for large-scale studies, encompassing substantial data and robust statistical power, becomes evident to capture these correlations and derive meaningful conclusions (5–7). Furthermore, paired analyses of high-throughput metagenomics and metabolomics data increase the discovery of molecular mechanisms behind reported associations between the microbiome and human health and disease (8). This approach also facilitates the discovery of biosynthetic pathways, with far-reaching implications for biotechnology and pharmaceutical industries (9). However, cross-contamination, throughput, and human error have been identified as major limitations when processing large numbers of samples (10–13). Sample plating and cell lysis using 96-well plates cause well-to-well contamination due to their wells having little separation and being connected by a single plate seal (10, 14). Recommendations to mitigate well-to-well effects include randomizing samples across plates, avoiding the processing of samples with different biomasses together, and opting for manual single-tube extractions (10, 12). However, implementing some of these recommendations comes with the cost of sacrificing throughput, increasing expenses, and introducing the potential for human error. For instance, single-tube extractions are more time-consuming than plate extractions due to their limited compatibility with automation (10). Computational methods are often used to remove contaminants (15–17); however, these methods have limitations: they either do not take into account well-to-well contamination or do not perform well under high levels of well-to-well contamination (15). Consequently, proactively preventing contamination is the preferable approach. Moreover, for study designs considering nucleic acid and metabolites, the process of separate extractions using technical replicates increases complexity, costs, and opportunities for technical effects and errors.

Here, we introduce a method for sample accession, DNA extraction, and metabolite extraction that preserves the high-throughput nature of plate-based extraction methods, while significantly reducing processing time and well-to-well contamination. Specifically, we perform metabolite extraction and cell lysis within single tubes to reduce well-to-well contamination. We utilize 1 mL barcoded tubes known as Matrix Tubes (catalog #3741; Thermo Fisher), which assemble into a rack of 96 tubes with a footprint fit for automation. These tubes serve as both collection and processing vessels and simplify sample accession as they are pre-barcoded and can be read in bulk. The tubes further remove the error-prone step of transferring samples from the collection vessel into wells of a 96-well plate. We use 95% (vol/vol) ethanol to stabilize the microbial community (18), which is also suitable as a solvent for metabolite extraction. To support these claims, we directly compared technical replicates using the Matrix method (i.e., use of Matrix Tubes) and a widely utilized plate-based method found in microbiome studies—the MagMAX Microbiome Ultra Nucleic Acid Isolation Kit (catalog #A42357; Thermo Fisher, MA, USA). This MagMAX kit has demonstrated superiority to other commercially available kits (19) and can be used for the Matrix method, with the exception of the bead plate (96-well plate) for lysis. To measure replicability and levels of well-to-well contamination, four laboratory technicians independently conducted each method in duplicate. Three technical replicates of human fecal samples, obtained from four volunteers, were collected under University of California San Diego's IRB #141853 for the comparison. Fecal swabs were transferred to corresponding positions in both 96-well plates and Matrix Tube racks, amounting to a total of 12 fecal swabs per method. Each swab was surrounded by 84 negative-control extraction blanks (Fig. 1) to observe well-to-well contamination; see the supplemental materials and methods. The plate-based method

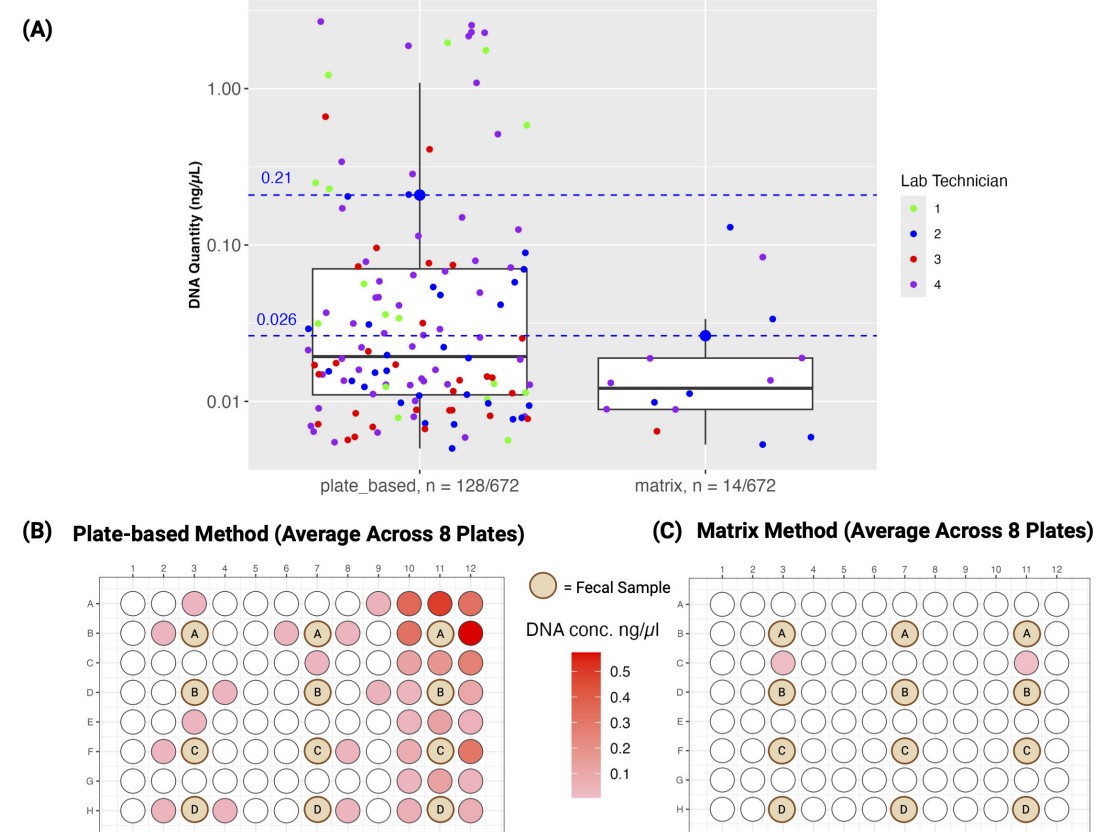

**FIG 1** 16S rRNA signal in blank samples. (A) Boxplot displaying blanks with DNA quantity > 0.005 ng/µL detected by qPCR for both plate-based and matrix extraction methods. Out of 672 blanks, 128 were contaminated during the plate-based method with an average concentration of 0.21 ng/µL and a median of 0.02 ng/µL, with only 14 contaminated during the Matrix method with an average concentration of 0.026 ng/µL and a median of 0.012 ng/µL. A Wilcoxon signed-rank test of plate-based vs Matrix blanks resulted in a significant difference of $W = 153{,}876$, $P < 2.2e{-}16$. The y-axis is displayed in a log10 scale. (B and C) A visual of the plate map for each method displaying the contaminated blank wells averaged across eight plates. The position of fecal samples is denoted by the brown circles with the subject ID.

adhered to the instruction manual. The Matrix method included a metabolite extraction step prior to nucleic acid extraction, where the samples were shaken in Matrix Tubes containing 95% (vol/vol) ethanol (Fig. 2). The resulting metabolite extracts were separated through centrifugation and transferred using a multichannel pipette into a 96-well plate suitable for mass spectrometry analysis (Fig. 2). We quantified each DNA extraction blank using quantitative polymerase chain reaction (qPCR) in triplicate reactions targeting the 16S rRNA gene. Our comparison of well-to-well contamination between the two extraction methods revealed that the Matrix method resulted in significantly less contamination compared to the plate-based method (Wilcoxon rank sum two-sided test; $W = 153{,}876$, $P < 2.2e{-}16$). The plate-based method revealed that 128 blanks out of 672 (19%) were contaminated (dsDNA quantity > 0.005 ng/µL) during processing with an average concentration of 0.21 ng/µL (Fig. 1B). The majority of contaminated blanks were located on the right side of the plate. We hypothesize that this is due to all four laboratory technicians being right-handed and therefore removing the seal from left to right. In contrast, the Matrix method had only 14 out of 672 blanks (2%) contaminated during processing with an average concentration of 0.026 ng/µL (Fig. 1C). We then conducted a comparison of DNA yields (ng/µL), to test whether differences were contributing to the contamination rate, but observed no statistically significant difference between the two methods (Fig. S1A). Laboratory technician and host subject had a greater influence on DNA yield (Fig. S1A), suggesting that variations in swabbing

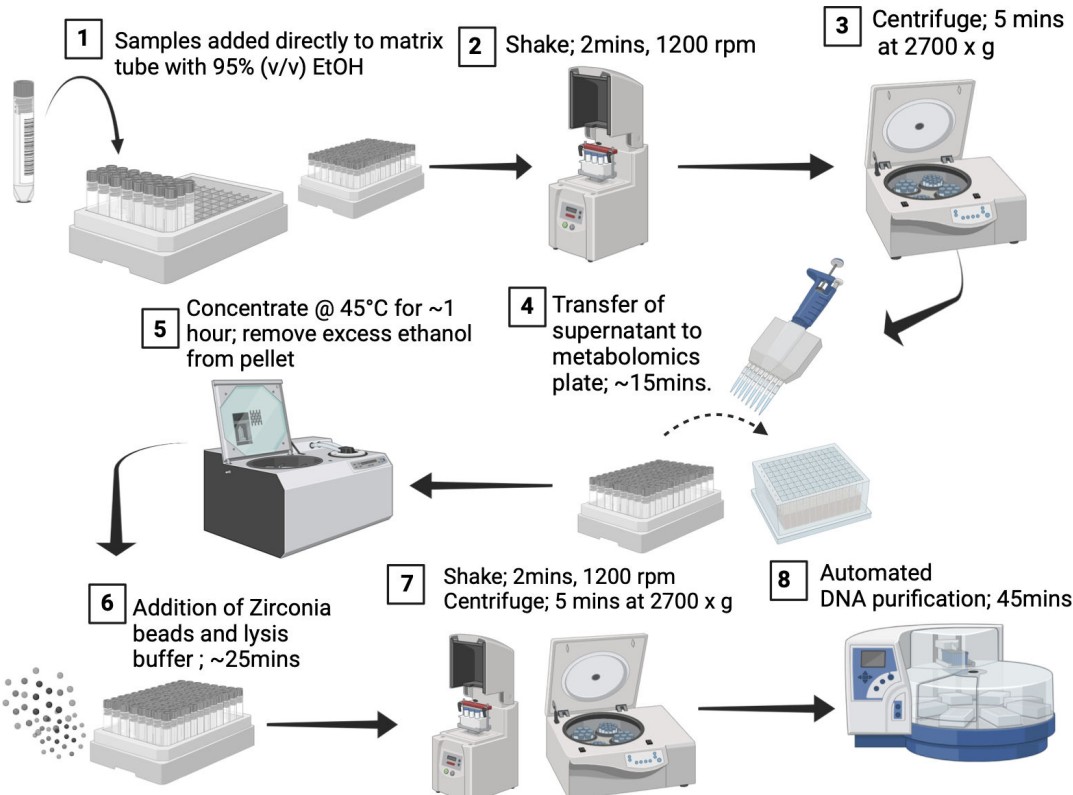

**FIG 2** Flowchart of Matrix method for sample accession, metabolite extraction, and DNA extraction. Step 1: samples are added directly into Matrix Tubes containing 400 µL of 95% (vol/vol) ethanol. Step 2: samples are homogenized using the SpexMiniG plate shaker (SPEXSamplePrep, NJ, USA) at 1,200 rpm for 2 minutes. Step 3: the Matrix rack is centrifuged for 5 min at 2,700 × g. Step 4: Matrix Tubes are de-capped by the automated instrument, Capit-All (Thermo Fisher Scientific, MA, USA). Supernatant is transferred from Matrix Tubes to corresponding wells of a 96-well metabolomics plate (catalog number: 75870-792, VWR), using a multi-channel pipette. Supernatant can later be analyzed by LC-MS/MS. Step 5: Excess ethanol is removed from matrix tubes using a SpeedVac (Thermo Fisher Scientific, MA, USA; catalog number: SPD2030A-220) set to 45°C for 60 min. Step 6: 30 µL of 0.1, 0.5, and 1 mm zirconia-silica beads are added to each matrix tube using the LabTie bead dispenser (MolGen). We add 600 µL of lysis buffer to each Matrix Tube. Matrix Tubes are capped by Capit-All. Step 7: bead beating is performed at 1,200 rpm for 2 min. The Matrix rack is centrifuged for 5 minutes at 2,700 × g. Step 8: Matrix Tubes are de-capped. Lysate is transferred from Matrix Tubes to corresponding wells of a 96-deep well plate using a multi-channel pipette. Nucleic acid purification is performed on the lysate using the KingFisher Flex according to the manufacturer's instructions of the MagMAX Microbiome Ultra Nucleic Acid Isolation Kit (Thermo Fisher, MA, USA). Created with BioRender.com.

techniques are likely attributable to differences among laboratory technicians, since they were not instructed to weigh each sample.

To compare microbial composition recovery between the Matrix method and the plate-based method, we extracted six technical replicates of human and mouse fecal samples from four subjects and six technical replicates of human saliva samples before and after brushing from three subjects using each method (IRB #141853 for feces and IRB #150275 for saliva). Mantel correlations reveal strong associations in microbial community beta-diversity for Jaccard, Canberra, and weighted and unweighted UniFrac distances across the two extraction protocols for both 16S and metagenomics data (Pearson's r > 0.77, P = 0.001) (Table S1). A visual of principal coordinate analysis of human fecal samples of weighted and unweighted UniFrac is shown in Fig. S1. We employed forward stepwise regression to assess the relative importance of factors influencing microbial community beta-diversity, analyzing unique distance metrics including weighted and unweighted UniFrac, Jaccard, and RPCA (20) (Table S2). This analysis demonstrated that, for all sample types, beta-diversity was predominantly influenced by host subject identity for both 16S and metagenomics data, with the extraction method having no significant impact on beta-diversity. Additionally, Faith's phylogenetic diversity (21)

(alpha-diversity) was also not significantly different between the two extraction methods (Mann-Whitney test, $P > 0.2$) (Table S3).

We compared metabolite recovery between the proposed Matrix method and a standard workflow for untargeted metabolomics analysis via LC-MS/MS of human fecal samples. Samples from three different subjects were extracted in triplicates using either the Matrix method (95% ethanol) or a 50% methanol extraction (22). Although clustering according to the extraction method could be observed, PCA revealed that host subject remained the strongest factor influencing clustering (PERMANOVA, subject, $R^2 = 0.47$, $F = 8.62$, $P < 0.001$) (Fig. S2A). Interestingly, 75% of the metabolic features recovered in the study could be observed through both extractions, which also included 95% of the annotated features using the GNPS spectral libraries (Fig. S2B). Additionally, the majority of the top 100 features discriminating subjects via pairwise supervised classification models (PLS-DA) obtained via 50% methanol extraction were also recovered and selected via the Matrix method (overlap ranging from 82% to 92%) (Fig. S2C).

We present a critical advance in sample handling that reduces the time required from technicians, decreases a well-known major source of contamination, and shortens the overall processing time for samples (Table S3). The Matrix method also enables paired nucleic acid and metabolomic assays from a single tubed sample. Our comparative analysis confirms that this hybrid approach of combining single tube extractions with 96-well plate magnetic-bead clean ups significantly reduces well-to-well contamination that occurs during plate-based methods. The incorporation of barcoded Matrix Tubes introduces a streamlined process for sample randomization, as automated plate readers such as the VisionMate (catalog #312800; Thermo Fisher Scientific) high-speed barcode reader can identify 96 samples on a plate simultaneously and connect the IDs and well coordinates to information in data management platforms. Once associated with sample metadata, capped tubes can be mixed and randomly assembled into a 96-tube rack. This randomization is crucial for mitigating bias during extractions or library prep, ensuring that any potential well-to-well contamination merely adds noise rather than bias to experimental designs. Furthermore, the substantial reduction in well-to-well contamination achieved by our Matrix method marks a pivotal advancement in microbiome research, helping to prevent contamination-related controversies (23, 24). Due to the ability of extracting both DNA and metabolites from a single tube and the elimination of the tedious, error-prone plating step, the Matrix method simplifies large sample size collection and metadata curation, reducing processing time by up to 50% (Table S3). Due to varying purchasing agreements across institutions, an accurate capital cost analysis cannot be provided. However, the Matrix method offers lower consumable costs (Table S4) and reduced sample loss, which help offset higher capital costs. Alternatively, to reduce capital costs, a handheld barcode scanner and an eight-channel screw-cap decapper can be used instead of the VisionMate barcode reader and Capit-All, respectively, although this will increase processing time. In the future, the Matrix method can be expanded to include additional modalities, such as RNA and protein, but further exploration of materials and methods is needed.

## ACKNOWLEDGMENTS

We thank Iveta Kalcheva from The Center for Microbiome Innovation for providing qPCR reagents.

This work was supported by The Alzheimer's Gut Microbiome Project, grant number U19AG063744.

## AUTHOR AFFILIATIONS

[1]Department of Pediatrics, University of California San Diego, La Jolla, California, USA
[2]Division of Biological Sciences, University of California San Diego, La Jolla, California, USA

³Collaborative Mass Spectrometry Innovation Center, Skaggs School of Pharmacy and Pharmaceutical Sciences, University of California San Diego, San Diego, California, USA

⁴Department of Computer Science and Engineering, University of California, San Diego, La Jolla, California, USA

⁵Department of Bioengineering, University of California, San Diego, La Jolla, California, USA

⁶Center for Microbiome Innovation, University of California, San Diego, La Jolla, California, USA

## AUTHOR ORCIDs

Caitriona Brennan ⓘ http://orcid.org/0000-0003-3943-6701
Rob Knight ⓘ http://orcid.org/0000-0002-0975-9019

## FUNDING

| Funder | Grant(s) | Author(s) |
| --- | --- | --- |
| HHS \| National Institutes of Health (NIH) | U19AG063744 | Caitriona Brennan |
| | | Pedro Belda-Ferre |
| | | Simone Zuffa |
| | | Vincent Charron-Lamoureux |
| | | Ipsita Mohanty |
| | | Gail Ackermann |
| | | Celeste Allaband |
| | | Madison Ambre |
| | | Tara Boyer |
| | | MacKenzie Bryant |
| | | Kalen Cantrell |
| | | Antonio Gonzalez |
| | | Daniel McDonald |
| | | Rodolfo A. Salido |
| | | Se Jin Song |
| | | Gillian Wright |
| | | Pieter C. Dorrestein |

## DATA AVAILABILITY

All sequencing data have been made publicly available at the EBI database (accession number PRJEB56784, ERP141755) and through Qiita (Qiita Study ID 14332). Mass spectrometry data generated in this study are available publicly in MassIVE under the accession number MSV000095260.

## ADDITIONAL FILES

The following material is available online.

### Supplemental Material

**Figure S1 (mSystems00985-24-s0001.tiff).** DNA yields and PCoA of fecal samples from different host subjects.

**Figure S2 (mSystems00985-24-s0002.tiff).** Comparison of metabolomics results between the 95% ethanol extraction (Matrix method) and 50% methanol.

**Supplemental material (mSystems00985-24-s0003.docx).** Figures S1 and S2, Tables S1 to S4, and supplemental materials and methods.

## Open Peer Review

**PEER REVIEW HISTORY (review-history.pdf).** An accounting of the reviewer comments and feedback.

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
