## [Reviewer comments · mSystems]

Clearing the Plate: A Strategic Approach to Mitigate Well-to-Well Contamination in Large-Scale Microbiome Studies

Caitriona Brennan, Pedro Belda-Ferre, Simone Zuffa, Vincent Charron-Lamoureux, Ipsita Mohanty, Gail Ackermann, Celeste Allaband, Madison Ambre, Tara Boyer, MacKenzie Bryant, Kalen Cantrell, Antonio González, Daniel McDonald, Rodolfo Salido, Se Jin Song, Gillian Wright, Pieter Dorrestein, and Rob Knight

Corresponding Author(s): Rob Knight, University of California San Diego

Review Timeline:

Submission Date:

July 19, 2024

Accepted:

August 6, 2024

Editor: Naseer Sangwan

Reviewer(s): Disclosure of reviewer identity is with reference to reviewer comments included in decision letter(s). The following individuals involved in review of your submission have agreed to reveal their identity: Jean Debedat (Reviewer #1)

Transaction Report:

DOI: <https://doi.org/10.1128/msystems.00985-24>

Re: mSystems00985-24 (Clearing the Plate: A Strategic Approach to Mitigate Well-to-Well Contamination in Large-Scale Microbiome Studies)

Dear Prof. Rob Knight:

Your manuscript has been accepted, and I am forwarding it to the ASM production staff for publication. Your paper will first be checked to make sure all elements meet the technical requirements. ASM staff will contact you if anything needs to be revised before copyediting and production can begin. Otherwise, you will be notified when your proofs are ready to be viewed.

Sincerely,
Naseer Sangwan
Editor
mSystems